# Inducing Reusable Skills From Demonstrations with Option-Controller Network

## Abstract

Humans can decompose previous experiences into skills and reuse them to enable fast learning in the future. Inspired by this process, we propose a new model called Option-Controller Network (OCN), which is a bi-level recurrent policy network composed of a high-level controller and a pool of low-level options. The options are disconnected from any task-specific information to model task-agnostic skills. The controller use options to solve a given task, and it calls one option at a time and waits until the option return. With the isolation of information and the synchronous calling mechanism, we can impose a division of works between the controller and options in an end-to-end training regime. In experiments, we first perform behavior cloning from unstructured demonstrations coming from different tasks. We then freeze the learned options and learn a new controller with an RL algorithm to solve a new task. Extensive results on discrete and continuous environments show that OCN can jointly learn to decompose unstructured demonstrations into skills and model each skill with separate options. The learned options provide a good temporal abstraction, allowing OCN to quickly transfer to tasks with a novel combination of learned skills *even with sparse reward*, while previous methods either suffer from the delayed reward problem due to the lack of temporal abstraction or a complicated option controlling mechanism that increases the complexity of exploration.

## 1 Introduction

Acquiring primitive skills from demonstrations and reusing them to solve a novel long-horizon task is a hallmark in human intelligence. For example, after learning the necessary skills (e.g., steering wheel, changing lanes) at a driving school, one could be capable of driving across the country by recombining the learned skills.

In hierarchical reinforcement learning, different temporal abstractions (Sutton et al., 1999; Dietterich, 2000; Parr & Russell, 1998; Nakanishi et al., 2004) are proposed to achieve structured exploration and transfer to a long-horizon task. However, when learning from scratch, these pure HRL methods share an *exploration problem*: it takes a significant amount of samples for random walk to induce a good temporal abstraction that leads to positive rewards at the beginning of training. To circumvent this issue, recent works (Le et al., 2018; Levy et al., 2018; Gupta et al., 2019; Jiang et al., 2019) have focused on learning useful skills in a pretraining phase first, and then reusing these skills when finetuning with HRL in the new environment. However, these methods either assume the existence of goal-conditioned policies or require intensive interaction with environments , which limits the practical values of these approaches. One general approach is to leverage the additional *unstructured demonstrations* during pretraining, e.g., compILE (Kipf et al., 2019) pretrains a VAE (Kingma & Welling, 2013) on the demonstrations and uses an action decoder for finetuning. Our work is in this line of research.

In this paper, we propose *Option-Control Network* (OCN). Inspired by the option framework (Sutton et al., 1999), OCN is a two-level recurrent policy network including a high-level controller and a pool of low-level options. At each time step, a selected low-level option outputs an action and a termination probability, and the high-level controller selects a new option whenever the old option is terminated. Inspired by Lu et al. (2021), we enforce a special recurrent hidden state updating rule to enforce the hierarchical constraint between the controller and the options so that the high-level

controller is updated less frequently than the low-level options while keeping the model end-to-end differentiable. As shown in Figure 1, OCN can jointly learn options and controllers with multitask behavior cloning from unstructured demonstrations. When given a new task, one could perform HRL finetuning by re-initializing the controller and freezing the options. This enables our model to generalize combinatorially to unforeseen conjunctions (Denil et al., 2017). Unlike previous works, our method does not require generative models (Eysenbach et al., 2018), goal-conditioned policies (Gupta et al., 2019), pre-specified policy sketch (Shiarlis et al., 2018) or constraints on the number of segments (Kipf et al., 2019), making our approach conceptually simple and general.

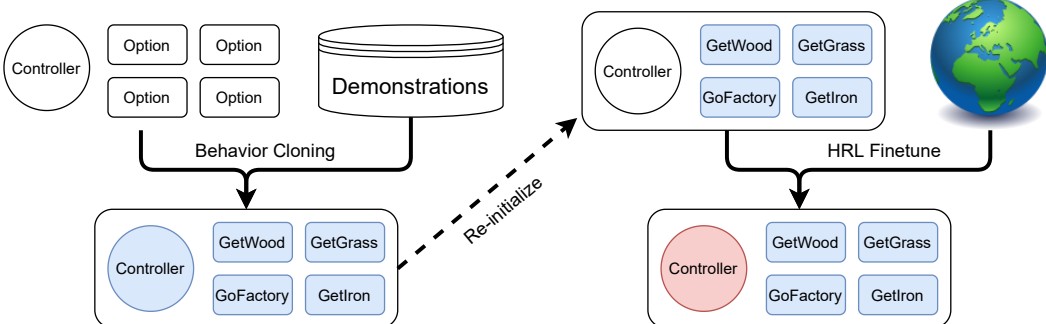

Figure 1: The training pipeline of OCN. Our model is composed of a controller (circle) and a options pool (rectangles). The controller and options are randomly initialized, which means each option does not correspond to a meaningful subtask. After behavior cloning, both options and controllers are induced (marked blue) and the options correspond to meaningful subtasks from demonstrations (e.g., get wood). Then we freeze the parameters in the options and re-initialize the controller. The controller is trained to adapt to the new environment with HRL (marked red).

We perform experiments in Craft (Andreas et al., 2017), a grid-world environment focusing on navigation and collecting objects, and Dial (Shiarlis et al., 2018), a robotic setting where a JACO 6DoF manipulator interact with a large number pad. Our results show that with unstructured demonstrations, OCN can jointly learn to segment the trajectories into meaningful skills as well as model this rich set of skills with our pool of low-level options. During HRL finetuning, we show that OCN achieves better performance in more complex long-horizon tasks with either sparse or dense reward compared with existing baselines. We also provide further visualization to show the discovered options are reused during finetuning. Our contributions can be concluded as:

1. We propose *Option-Controller Network*, a bi-level recurrent policy network that uses a special recurrent hidden state updating rule to enforce the hierarchical constraints.

2. We show that the OCN can discover the optimum options decomposition from unstructured demonstration via multitasking behavior cloning.

3. We also demonstrate that the learned options can be coupled with a new controller to solve an unseen long-horizon task via interaction with the environment.

## 2 RELATED WORK

Learning to solve temporally extended tasks is an important question for Hierarchical Reinforcement Learning (HRL), including option frameworks (Sutton et al., 1999), HAM (Parr & Russell, 1998) and max-Q (Dietterich, 2000). With the popularity of neural nets, recent works propose to use a bi-level neural network such as option critics (Bacon et al., 2017), feudal networks (Vezhnevets et al., 2017), generative models with latents (Nachum et al., 2018), and modulated networks (Pashevich et al., 2018). These models can be furthered combined with hindsight memory (Levy et al., 2018) to increase the sample efficiency. Our work can also be viewed as designing a specific neural architecture for HRL.

However, as discussed in Section 1, a pure HRL method suffers from serious exploration challenges when learning from scratch (Gupta et al., 2019). A general approach to tackle this problem is to introduce a pretraining phase to "warm up" the policy. Recent works propose to pretrain the policy

with an intrinsic diversity reward (Eysenbach et al., 2018) or language abstraction (Jiang et al., 2019), which is shown to be useful in the HRL. However, assuming access to an environment in the pretrain phase might be infeasible in many tasks. A principal approach is to leverage the additional unstructured expert demonstrations and performs imitation learning during pretraining. Our work can be viewed as solving the "cold start" problem for the option framework (Sutton et al., 1999).

Recent works build upon this "imitation - finetune" paradigm. With the prevalence of goal-conditioned policies (Schaul et al., 2015; Kaelbling, 1993; Levy et al., 2018) in robotics, these methods leverage demonstrations with relabelling technique to pretrain the low-level policy (Gupta et al., 2019) or a generative model (Lynch et al., 2020). However, they exploit the fact that the ending state of a trajectory segment can be described as a point in the goal space. Hence it is difficult to apply them beyond goal-conditioned policies. CompILE (Kipf et al., 2019) treats the segment boundaries as latent variables, and their model can be trained end-to-end with soft trajectory masking. However, CompILE requires specifying the number of segments, while OCN only requires that the number of options to be large enough (Appendix A.5. Following a similar VAE setting, SPiRL (Pertsch et al., 2020) includes a skill prior and a skill autoencoder. They encoder skills into latent variables and use a skill prior to generate high-level instructions – a sequence of latent variables. However, SPiRL define skills as a sequence of actions with a fix horizon $H$, which could prevent the model from learning skills with clear semantics. Modular policy networks (Andreas et al., 2017; Shiarlis et al., 2018) are also used in this paradigm, where each subtask corresponds to a single modular policy. However, in this setting, the demonstration needs to be segmented beforehand, which requires additional human labor. On the contrary, our work focused on using unstructured demonstrations. OptionGAN (Henderson et al., 2018) proposes a Mixture-of-Expert (MoE) formulation and performs IRL on the demonstration. However, without an explicit termination function, the learnt expert networks do not provide time-extended actions for the high-level controller. As a result, this method still suffers from problems of exploration with sparse rewards (as also seen in our experimental comparison with an MoE baseline).

Extracting meaningful trajectory segments from the unstructured demonstration is the focus of Hierarchical Imitation Learning (HIL). These works can be summarized as finding the optimal behavior hierarchy so that the behavior can be better predicted (Solway et al., 2014). DDO (Fox et al., 2017) proposes an iterative EM-like algorithm to discover multiple levels of options, and it is applied in the continuous action space (Krishnan et al., 2017) and program modelling (Fox et al., 2018). VALOR (Achiam et al., 2018) extends this idea by incorporating powerful inference methods like VAE (Kingma & Welling, 2013). Directed-Info GAIL (Sharma et al., 2018) extracts meaning segments by maximizing the mutual information between the subtask latent variables and the generated trajectory. Ordered Memory Policy Network (OMPN) (Lu et al., 2021) proposes a hierarchical inductive bias to infer the skill boundaries. The above works mainly focus on skill extraction, so it is unclear how to use the segmented skills for RL finetuning. Although OCN shares a similar inductive bias with OMPN, OCN replaces the continuous hidden states communication with a softmax distribution over multiple low-level modules (options). This enables OCN to model different subtasks with different options and to effectively reuse them in a new task.

## 3 METHODOLOGY

An Option-Controller Network (OCN) includes a set of $N$ options $\{\mathbf{o}_1, ..., \mathbf{o}_N\}$ and a controller $\mathbf{c}$. As shown in figure 1, the OCN starts by using the controller to choose an option to execute the first subtask. Once the subtask is done, the controller will choose another option to execute the second subtask, and so on, until the goal of the task is achieved. OCN shares a similar inductive bias with Ordered Memory Policy Network (OMPN) (Lu et al., 2021), that the lower-level components (options) execute more frequently than the higher-level components (controllers). The inductive bias enables OCN to induce the temporal hierarchical structure of unstructured demonstrations.

### 3.1 OPTION AND CONTROLLER

**Option** As shown in the middle of Figure 2, an option $\mathbf{o}_i$ models a skill that can solve one specific subtask, for example *get wood*, *get iron* or *make at workbench*. It can be described as:

$$\mathbf{p}_{i,t}^{\mathbf{o}}, \mathbf{h}_{i,t}^{\mathbf{o}}, e_{i,t} = \mathbf{o}_i(\mathbf{x}_t, \mathbf{h}_{i,t-1}^{\mathbf{o}}) \tag{1}$$

where $\mathbf{x}_t$ is the observation at time step $t$, and $\mathbf{h}^o_{i,t-1}$ is the hidden state of the respective option at time step $t-1$; $\mathbf{h}^o_{i,t}$ is the hidden state of $\mathbf{o}_i$ at time step $t$; $e_{i,t}$ is a scalar between 0 and 1, represents the probability that the current option is done; $\mathbf{p}^o_{i,t}$ is a distribution of actions, including *move up*, *move left* and *use*. These actions are the smallest elementary operations that an agent can execute. During the execution of an option, if probability $e_{i,t}$ is 0, the option will keep executing the current subtask; if $e_{i,t}$ is 1, the option will stop the execution and return to the controller for the next subtask. In our work, each option maintains a separate set of parameters.

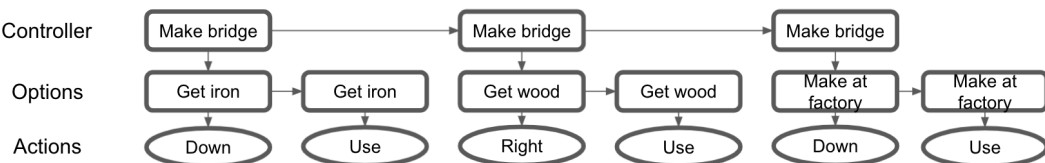

Figure 2: An example of OCN. The controller $\mathbf{c}$ models the task *make bridge*. Three options separately model subtasks *get iron*, *get wood* or *make at factory*.

**Controller**   As shown at the top of Figure 2, a controller $\mathbf{c}$ models a higher level task, like *make bed*, *make axe*, or *make bridge*. Each of these tasks can be decompose to a sequence of subtasks. For example, *make bridge* can be decompose to 3 steps: 1) *get iron*, 2) *get wood*, 3) *make at factory*. Thus a controller can also be represented as:

$$\mathbf{p}^c_t, \mathbf{h}^c_t, e^c_t = \mathbf{c}(\mathbf{x}_t, \mathbf{h}^c_{t-1}) \tag{2}$$

where $\mathbf{p}^c_t$ is a distribution over the set of options $\{\mathbf{o}_i\}$, $\mathbf{h}^c_t$ is the hidden state for controller, $e^c_t$ is the probability that the current task is done. In this OCN architecture, we don't need the $e^c_t$, since the environment will provide signal(reward) once the task is done. However, OCN can be easily expanded to a multi-level model. In this multi-levels model, a set of multiple controllers become options for a higher-level controller, and their respective tasks become subtasks for a more complicated task.

**Cell Network**   In OCN, options and controllers share the same format for input and output. Thus, we parameterize them with the same neural network architecture. To model the policy of controllers and options, we proposed the following cell network:

$$\hat{h}_t = \text{MLP}\left([x_t, h_{t-1}]\right) \tag{3}$$

$$p_t = \text{softmax}(\mathbf{W}_{\text{act}}\hat{h}_t + \mathbf{b}_{\text{act}}) \tag{4}$$

$$h_t = \tanh(\mathbf{W}_{\text{hid}}\hat{h}_t + \mathbf{b}_{\text{hid}}) \tag{5}$$

$$e_t = \text{sigmoid}(\mathbf{w}_{\text{end}}\hat{h}_t + b_{\text{end}}) \tag{6}$$

$x_t$ is the raw observation, the shape of the vector depends on the environment. $h_{t-1}$ is the recurrent hidden state of size $d_{\text{hid}}$, it allows the model to remember important information from previous time steps. MLP is a multi-layer neural network of Depth $l_{\text{MLP}}$ and hidden size $d_{\text{MLP}}$. We use $\tanh$ as activation function for MLP. $\hat{h}_t$ is a vector of size $d_{\text{MLP}}$. $\mathbf{W}_{\text{act}}$ is a matrix of size $n_{\text{act}} \times d_{\text{MLP}}$, where $n_{\text{act}}$ is number of actions. $\mathbf{W}_{\text{hid}}$ is a matrix of size $d_{\text{hid}} \times d_{\text{MLP}}$. $\mathbf{w}_{\text{end}}$ is a vector of size $d_{\text{MLP}}$. Following the fast and slow learning idea proposed in Madan et al. (2021), we introduce a temperature term $T$ to controller's softmax function:

$$p^c_t = \text{softmax}\left(\frac{\mathbf{W}_{\text{act}}\hat{h}_t + \mathbf{b}_{\text{act}}}{T}\right) \tag{7}$$

A large temperature $T$ allows the option to output smoother distribution at the beginning of training. It also reduces the scale of gradient backpropagated into the controller. This results in the controller changes and updates slower than options. We found $T$ makes OCN become more stable in imitation learning and converge to a better hierarchical structure.

### 3.2   OPTION-CONTROLLER FRAMEWORK

Given the definition for options and controllers, we can further formulate OCN. As shown in Figure 3a, at the first time step, the controller computes a probability distribution over options for the first

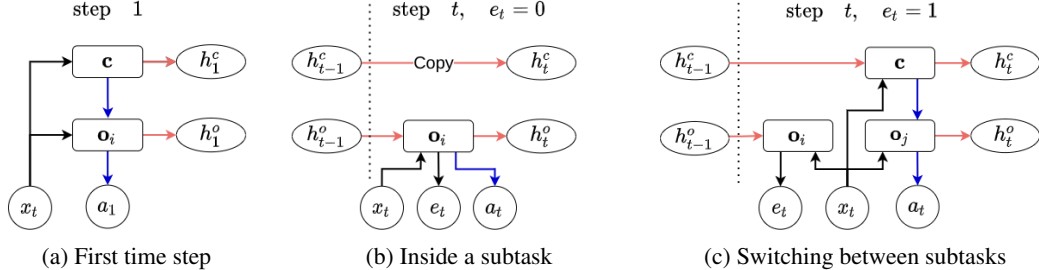

| step 1 | step $t$, $e_t = 0$ | step $t$, $e_t = 1$ |
| (a) First time step | (b) Inside a subtask | (c) Switching between subtasks |

Figure 3: The three different phase of OCN: (a) At the first time step, the controller selects an option $\mathbf{o}_i$; The option $\mathbf{o}_i$ outputs the first action $\mathbf{a}_1$. (b) If the previous option $\mathbf{o}_i$ predict that the subtask is not finish; The option $\mathbf{o}_i$ then continue outputs action $\mathbf{a}_t$; The controller hidden state is copied from previous time step. (c) If the previous option $\mathbf{o}_i$ predict that the subtask is done; The controller then selects a new option $\mathbf{o}_j$ and updates the controller hidden state; The new option $\mathbf{o}_j$ outputs action $\mathbf{a}_t$. Blue arrows represent probability distributions output by controller and options. Red arrows represent recurrent hidden states between time steps.

subtask, and options execute their first steps:

$$\mathbf{p}_1^{\mathbf{c}}, \mathbf{h}_1^{\mathbf{c}} = \mathbf{c}(\mathbf{x}_1, \mathbf{h}_0^{\mathbf{c}}) \tag{8}$$

$$\mathbf{p}_{i,1}^{\mathbf{o}}, \mathbf{h}_{i,1}^{\mathbf{o}}, e_{i,1} = \mathbf{o}_i(\mathbf{x}_1, \mathbf{h}_{i,0}^{\mathbf{o}}) \tag{9}$$

$$\mathbf{p}_1^{\mathbf{a}} = \sum_i p_{1,i}^{\mathbf{c}} \mathbf{p}_{i,1}^{\mathbf{o}} \tag{10}$$

where $h_0^{\mathbf{c}}$ and $h_{i,0}^{\mathbf{o}}$ are initial hidden states for controller and options, $p_1^{\mathbf{a}}$ is a distribution for actions. The output $p_1^{\mathbf{a}}$ is formulated as a mixture of experts, where experts are options and the gating model is the controller.

At time steps $t > 1$, the options first execute one step to decide whether this subtask is done. If the subtask is unfinished, the option then outputs an action distribution, as shown in Figure 3b:

$$\hat{\mathbf{p}}_{i,t}^{\mathbf{o}}, \hat{\mathbf{h}}_{i,t}^{\mathbf{o}}, e_{i,t} = \mathbf{o}_i(\mathbf{x}_t, \mathbf{h}_{i,t-1}^{\mathbf{o}}) \tag{11}$$

$$e_t = \sum_i p_{t-1,i}^{\mathbf{c}} e_{i,t} \tag{12}$$

$$\hat{\mathbf{p}}_{i,t}^{\mathbf{a}} = \sum_i p_{t-1,i}^{\mathbf{c}} \hat{\mathbf{p}}_{i,t}^{\mathbf{o}} \tag{13}$$

where $e_t$ is the probability that the previous subtask is done and $\hat{\mathbf{p}}_{i,t}^{\mathbf{a}}$ is the action distribution if the subtask is not done. If the previous subtask is done, the controller $\mathbf{c}$ need to select a new option distribution for the next subtask and reinitialize the option, as shown in Figure 3c:

$$\mathbf{p}_t^{\prime\mathbf{c}}, \mathbf{h}_t^{\prime\mathbf{c}} = \mathbf{c}(\mathbf{x}_t, \mathbf{h}_{t-1}^{\mathbf{c}}) \tag{14}$$

$$\mathbf{p}_{i,t}^{\prime\mathbf{o}}, \mathbf{h}_{i,t}^{\prime\mathbf{o}}, e_{i,t}' = \mathbf{o}_i(\mathbf{x}_t, \mathbf{h}_{i,0}^{\mathbf{o}}) \tag{15}$$

$$\mathbf{p}_t^{\prime\mathbf{a}} = \sum_i p_{t,i}^{\prime\mathbf{c}} \mathbf{p}_{i,t}^{\prime\mathbf{o}} \tag{16}$$

where $\mathbf{h}_t^{\prime\mathbf{c}}$, $\mathbf{h}_t^{\prime\mathbf{c}}$, $\mathbf{p}_t^{\prime\mathbf{c}}$ and $p_{t,i}^{\prime\mathbf{c}}$ are hidden states and distributions for the next subtask if the previous subtask is done. Thus, we can formulate the output at time step $t$ as a weighted sum of the two situations:

$$\begin{bmatrix} \mathbf{h}_t^{\mathbf{c}} \\ \mathbf{p}_t^{\mathbf{c}} \\ \mathbf{h}_t^{\mathbf{o}} \\ \mathbf{p}_t^{\mathbf{a}} \end{bmatrix} = e_t \begin{bmatrix} \mathbf{h}_t^{\prime\mathbf{c}} \\ \mathbf{p}_t^{\prime\mathbf{c}} \\ \mathbf{h}_t^{\prime\mathbf{o}} \\ \mathbf{p}_t^{\prime\mathbf{a}} \end{bmatrix} + (1 - e_t) \begin{bmatrix} \mathbf{h}_{t-1}^{\mathbf{c}} \\ \mathbf{p}_{t-1}^{\mathbf{c}} \\ \hat{\mathbf{h}}_t^{\mathbf{o}} \\ \hat{\mathbf{p}}_t^{\mathbf{a}} \end{bmatrix} \tag{17}$$

The equation 17 provides OCN an internal hierarchical inductive bias, that a higher-level component ($\mathbf{c}$) only update its recurrent hidden state and output a new command ($\mathbf{p}_t^{\prime\mathbf{c}}$) when its current functioning subordinate ($\mathbf{o}_i$) reports "done".

### 3.3 INDUCING AND REUSING SKILLS

**Imitation Learning and Inducing Skills**  OCN imitates and induces skills from unstructured demonstrations. In the rest of this paper, $\mathbf{d}$ represents an unstructured demonstration $\{(x_t, a_t)\}_{t=1}^{T}$ $\mathbf{D}$ represents a set of demonstrations of different tasks $[(\mathbf{d}_1, \tau_1), (\mathbf{d}_2, \tau_2), ...]$, where $\tau_i$ are task ids, belongs to a shared task set $\mathbf{T}$.

Given a demonstration $\mathbf{d}$, OCN can perform behavior cloning with a negative log-likelihood loss:

$$loss = \text{average}_t \left( \text{NLLLoss}(\mathbf{p}_t^{\mathbf{a}}, a_t) \right) \tag{18}$$

For different tasks $\tau$, we can use two different methods to model their associated controllers. The first method is to assign one controller $\mathbf{c}_\tau$ and a initial hidden state $\mathbf{h}_{\tau,0}^{\mathbf{c}}$ to each $\tau$. The second method is to share the controller $\mathbf{c}$, but assign a different initial hidden state $\mathbf{h}_{\tau,0}^{\mathbf{c}}$ to each $\tau$. We choose the second method in this work because sharing $\mathbf{c}$ could avoid the risk that different controllers choose to model the same subtask with options. During the imitation learning, OCN allows gradient backpropagation through all probabilities $\mathbf{p}$. Thus, the gradient descent will try to induce an optimal set of options that can best increase the likelihood of the data.

**Reinforcement Learning and Reusing Skills**  Given the induced options from imitation learning, our model can learn to solve a new task by reusing these skills via reinforcement learning. For example, after training on demonstrations of task 1 and task 2, OCN induce $N$ options $\{\mathbf{o}_1, ..., \mathbf{o}_N\}$. Given a new task 3 without demonstrations, we can initialize a new controller $\mathbf{c}_3$, that takes observations as input and outputs a probability distribution over $N$ induced options. To learn $\mathbf{c}_3$, we freeze all options and use PPO (Schulman et al., 2017) algorithm to learn $\mathbf{c}_3$ from interactions with the environment. During the training, once the controller outputs an option distribution $p^{\mathbf{c}}$, OCN samples from the distribution, the sampled option will rollout until it's done, then the process will repeat until the task is solved. Thus, in the RL phase, our model only needs to explore at options space, which significantly reduces the number of interaction steps to solve the new tasks. We outline the process in Appendix A.1.

## 4 EXPERIMENTS

We test OCN in two environments. For the discrete action space, we use a grid world environment called **Craft** adapted from Andreas et al. (2017). For the continuous action space, we have a robotic setting called **Dial** (Shiarlis et al., 2018), where a JACO 6DoF manipulator interacts with a number pad. We compare OCN with three baselines including task decomposition methods and hierarchical methods: (1) compILE (Kipf et al., 2019), which leverages Variational Auto-Encoder to recover the subtask boundaries and models the subtasks with different options. (2) OMPN (Lu et al., 2021) which studies inductive bias and discovers hierarchical structure from demonstrations. (3) Mixture-of-Experts (MOE), which uses a similar architecture as OCN, but the controller predicts a new distribution over options at every time step. This baseline is designed following the MoE framework proposed in Henderson et al. (2018). Implementation details for OCN and baselines can be found in Appendix A.2 and A.3.

### 4.1 CRAFT

In this environment, an agent can move in a 2D grid map with actions (*up*, *down*, *left*, *right*) and interact with the objects with the action *use*. The environment includes 4 subtasks: A) *get wood*, B) *get gold*, C) *get iron*, D) *get grass*. They require the agent to locate and collect a specific type of object. For example, subtask A, *get wood*, requires the agent to first navigate to the block that contains wood, then execute a *use* action to collect one unit of wood. A task requires the agent to finish a sequence of subtasks in the given order. The environment can provide either sparse reward or dense reward. In the dense reward, the agent receives rewards after completing each subtask while in the sparse reward, the agent only receives rewards after completing all subtasks.

**S1: Transferring from Single Agent**  In this setting, the training task set is $\{\text{AC}, \text{CD}, \text{DA}\}$. During the imitation phase, we pretrain an OCN with one controller $\mathbf{c}_1$ and three options $\{\mathbf{o}_1, \mathbf{o}_2, \mathbf{o}_3\}$ to imitate these demonstrations. During the fine-tuning phase, the model needs to solve three new tasks: $\{\text{ADC}, \text{CAD}, \text{DCA}\}$. We initialize a new controller for each while freezing the parameters

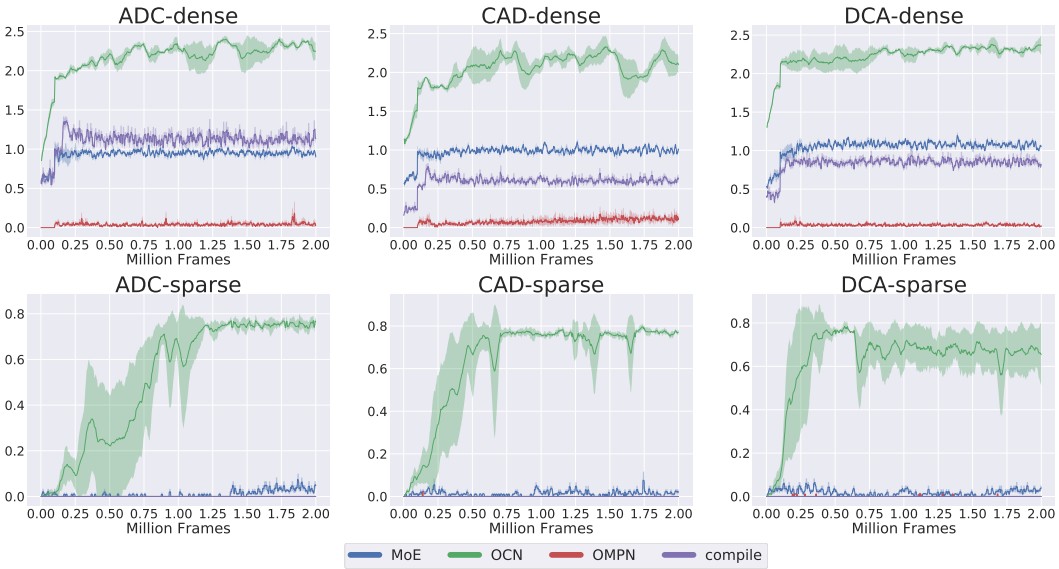

Figure 4: The learning curve of different methods on three finetuning tasks of **S1**. `dense` means dense reward setting. `sparse` means sparse reward setting.

of options. This is the classical setting where an agent is required to learn skills from short expert demonstrations and to transfer to long-horizon tasks.

As is shown in Figure 4, our method converges faster and achieves higher performance than baselines in both dense and sparse reward settings. With dense rewards, our method achieves double the returns than the strongest baseline. In the sparse reward setting, our method can get an average return of 0.7 with the maximum being 1, while other baselines struggle. We find that MoE fails to achieve similar performance even with a very similar architecture as OCN. The only difference is that MoE does not model the termination of an option and the controller selects a new option every time step. This result shows that exploring in option space is more efficient than other schemes, provided the new task is expressible as a combination of previous observed subtasks.

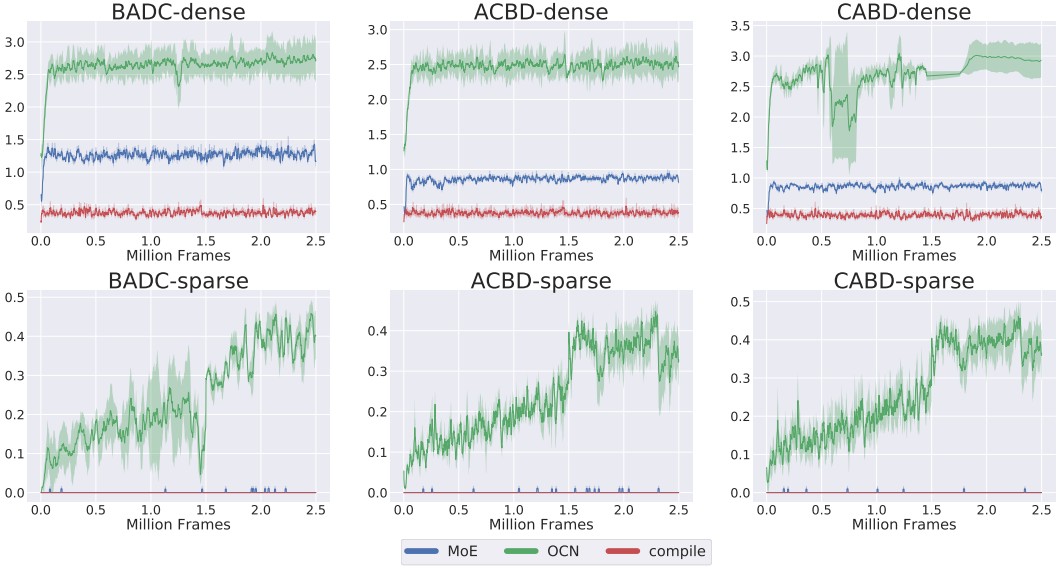

Figure 5: The learning curve of different methods on three finetuning tasks of **S2**. OMPN is not included because it does not learn an explicit set of options.

**S2: Transferring from Multiple Agents** In this setting, we have two disjoint task sets. The first set is {AB, BA} and the second task set is {CD, DC}. We train two separate OCN models. Each model includes a controller and two options. Thus, at the end of imitation phase, we obtain four options {$o_1, ..., o_4$}. Then we initialize three new controllers to solve three new tasks: {BADC, ACBD, CABD}.

This setting is related to the problem of data islands and federated learning (Yang et al., 2019), where two companies could each pretrain models on their separate datasets, merge the induced options, and share the controller finetuned on more challenging tasks. This is made possible because of the highly *modularized design* of our architecture.

The results are shown in Figure 5. We show that OCN can still reuse merged option pools, while other baseline methods fail at this setting. CompILE uses a continuous latent variable for communication between the controller and the action decoder, which causes compatibility issues while merging skills from different models. The MoE method still suffers from the long horizon problem. Overall, this result highlights the flexibility of OCN and its promise in maintaining data privacy for collaborative machine learning applications.

## 4.2 DIAL

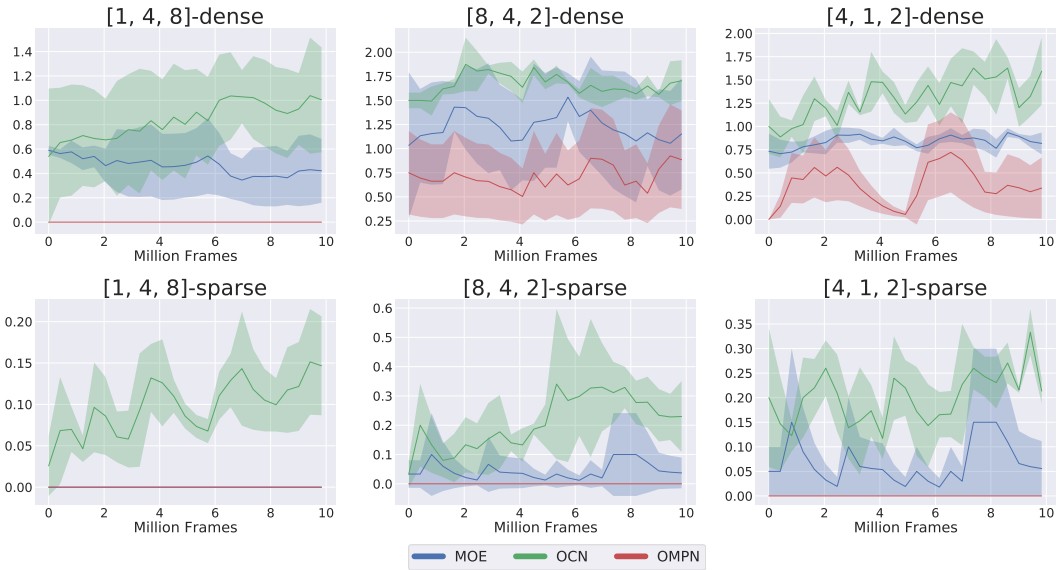

Figure 6: The learning curve of different methods on three finetuning tasks of Dial. `dense` means dense reward setting. `sparse` means sparse reward setting.

In this experiment, the task requires the agent to move the robotic arm to dial a PIN – a sequence of digits that should be pressed in the given order. We choose a set $S = (1, 2, 4, 8)$ of 4 digits as the environment. Demonstrations contain state-action trajectories that the robotic arm press a 2-digits pin randomly sampled from $S$. Demonstrations are generated from a PID controller that moves the robotic arm to predefined joint angles for each digit. The fine-tuning task is dialing a 3-digits PIN sampled from $S$. The environment can provide either sparse rewards or dense rewards. In the dense reward setting, the agent can receive a reward after pressing each digit in the PIN. In the sparse reward setting, the agent can only receive a reward if all the digits in the PIN are pressed. The state has 39-dimensions which containing the 9 joint angles and the distance from each digit on the dial pad to the finger of the robotic arm in three dimensions. We compare OCN with two baseline methods: OMPN and MoE.

Figure 6 shows the learning curve of three different fine-tuning tasks. We find that OCN consistently outperforms baseline methods. It's worth noting that, although OCN is designed to learn a discrete transition between subtasks, it is still capable of providing a good solution for continuous settings.

### 4.3 MODEL ANALYSIS

**Visualization** Appendix A.6 shows several trajectories produced by OCN, including imitation learning trajectories and reinforcement learning trajectories. As shown in Figure 10 and 11, OCN can accurately segment the demonstration into different subtasks and unambiguously associate options with subtasks. At the reinforcement learning phase, the discovered options are reused to solve their assigned tasks when coupled with a new controller to solve a long horizon task.

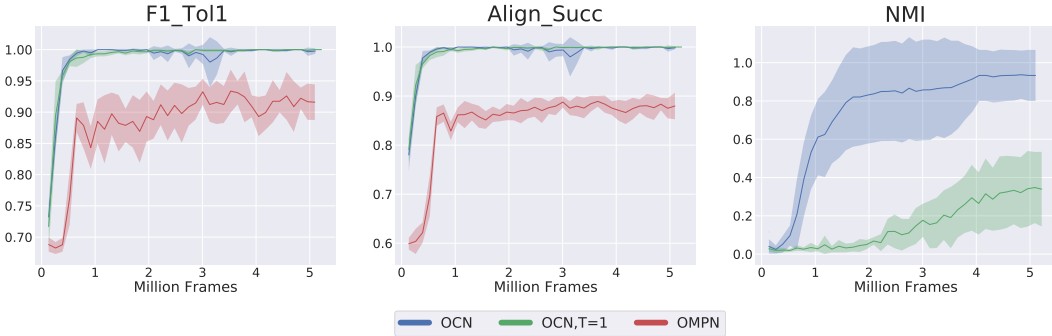

Figure 7: Comparison of unsupervised trajectory parsing results during the imitation phase with OMPN (Lu et al., 2021). The F1 score with tolerance (**Left**) and Task Alignment (**Center**) show the quality of learned task boundaries. The normalized mutual information (**Right**) between the emerged option selection $\mathbf{p}_t^c$ and the ground-truth shows that OCN learns to associate each option to one subtask. `T=1` means that the temperature term in the controller is removed.

**Quantitative Analysis** Figure 7 shows the performances of parsing and option-subtask correlation during the imitation phase. We find that OCN can converge faster and achieve better parsing performance than the OMPN model. The NMI figure in Figure 7 shows that, during the imitation phase, randomly initialized options slowly converged to model different subtasks. At the end of imitation, OCN shows strong alignment between options and subtasks. In 4 out of 5 runs, OCN actually achieves NMI=1, which means that the alignment between

Table 1: The success rate of each option when testing on different subtasks.

| Option | subtask | A | C | D |
|---|---|---|---|---|
| 1 | | 0.96 | 0.07 | 0.03 |
| 2 | | 0.00 | 0.02 | 0.95 |
| 3 | | 0.01 | 0.98 | 0.01 |

option and subtask is perfect. On the other hand, if we remove the temperature term (i.e. set $T = 1$) in the controller, the NMI drops significantly. This result suggests that the fast and slow learning schema is important for the model to learn the correct alignment between options and subtasks. Furthermore, Table 1 shows the success rate of using each option to solve each subtask. We find that there is a one-to-one correspondence between subtasks and learned options.

In the Appendix A.5, we show that OCN is not sensitive to the number of options $N$, as long as it's larger than the intrinsic number of skills in demonstrations. If $N$ is too small, then the performance of skill transfer may decrease. So it's always recommended to choose a larger $N$.

## 5 CONCLUSION

In this paper, we proposed a novel framework: Option-Controller Network(OCN). It is composed of a controller and a set of options. After training on a set of demonstrations, OCN can automatically discover the temporal hierarchical structure of training tasks and assign different options to solve different subtasks in the hierarchy. Experiment results show that our model can effectively associate subtasks with options. And a newly initialized controller can leverage previously learned options to solve complicated long horizon tasks via interaction with the environment. In this finetuning process, OCN shows strong combinatorial generalization. It outperforms previous baseline methods by a large margin. Overall, this method provides a simple and effective framework for hierarchical imitation learning and reinforcement learning in discrete space.

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

## A APPENDIX

### A.1 REINFORCEMENT LEARNING ALGORITHM

---
**Algorithm 1:** PPO, Adapt OCN to a new task

---
Initialize controller $\mathbf{c}$;
Freeze all options $\{\mathbf{o}_{1...N}\}$;
**for** *iterations=1,2,...* **do**
    **for** *actor=1,2,...* **do**
        **for** *step t=1,2,...,T* **do**
            $p^{\mathbf{c}} = \mathbf{c}(x_t, h^{\mathbf{c}}_{t-1})$;
            $i = \text{sample}(p^{\mathbf{c}})$;
            Rollout $\mathbf{o}_i$ until sample$(e_i) = 1$;
        **end**
        Compute advantage estimates $\hat{A}_1, ..., \hat{A}_T$;
    **end**
    Optimize surrogate $L$ wrt $\mathbf{c}$, with $K$ epochs and minibatch size $B$;
**end**

---

### A.2 IMPLEMENTATION DETAILS

We train all imitation learning methods by utilizing behaviour cloning with a batch size of 512 and a learning rate of 0.001. For each task, we sample 6000 demonstrations and split 80% for training and 20% for validation. For reinforcement learning, we use PPO algorithm with a batch size of 1024 and a learning rate of 0.0003. We use Adam optimizer and a linear schedule to adjust the learning rate. The hidden state size $d_{\text{hid}}$ of OCN and baselines is 128. The depth $l_{\text{MLP}}$ of cell network is 2. The temperature $T$ for controller's softmax function is 10. The hyparameters used in IL and RL are listed in table 2 and 3. We perform imitation learning and reinforcement learning phase with a Tesla V100 GPU.

Table 2: Imitation Learning Parameters

| Hyparameter | Value |
|---|---|
| batch size | 512 |
| learning rate | 0.001 |
| train episodes | 1500 |
| hidden size | 128 |
| optimizer | Adam |
| temperature for Softmax | 10 |

Table 3: Reinforcement Learning Parameters

| Hyparameter | Value |
|---|---|
| batch size | 256 |
| learning rate | 0.0003 |
| hidden size | 128 |
| entropy | 0.001 |
| ppo epoch | 4 |
| gamma | 0.97 |
| optimizer | Adam |

### A.3 BASELINES

**compILE** We modify the encoder and decoder of compILE so that the model can adapt to the observation of our environment. We use the discrete latent as the original paper describes. We re-initialize the encoder to predict latent for the new tasks and freeze the decoder which predicts the actions in RL.

**MoE** This baseline has a similar architecture with OCN, but it doesn't have a terminate signal and has to predict options distribution and select options at each step. We pretrain MoE with a controller and options in the imitation phase and re-initialize a new controller for the new tasks while freezing options as we do with OCN.

**OMPN** We can directly use this baseline in the imitation phase. However, since OMPN has strong connections(e.g., bottom-up and top-down recurrence) between the higher-level model and lower-level model, we don't re-initialize the higher-level model in RL.

### A.4 TASK ALIGNMENT EVALUATION

We use three metrics to evaluate the performance of parsing and option-subtask correlation: a) task align accuracy, b) F1 scores with tolerance and c) normalized mutual information scores(NMI). The first two metrics we use are the same as those defined in OMPN. The NMI is a normalization of the mutual information score between two clusterings, which are subtasks and options in our experiment. We give the formulation of NMI between two clusterings $U, V$ as:

$$NMI(U,V) = \frac{MI(U,V)}{mean(H(U),H(V))}$$

where $MI(U,V)$ is the mutual information and $H$ is the entropy. The values of $NMI$ is from 0 to 1. The higher values mean the higher correlation between subtasks and options.

### A.5 HYPERPARAMETERS ANALYSIS

Our model does not require the assumption about the number of skills. We analyze the effect of the number of options $K$. As shown in Figure 8, when $K$ is larger than or equal to the number of skills, which is 3 in this experiment, our model basically remains similar results at three metrics: Align Acc, F1 Tol1, and NMI and achieve almost 1. When $K = 2$, which means $K$ is smaller than the number of skills, one of the options must execute two different skills, which is contrary to our assumption and only achieves 0.4 at $NMI$. We also compare the prediction accuracy of the actions and the returns in Figure 9. Our performance isn't influenced by the number of skills when $K$ is larger than the number of skills.

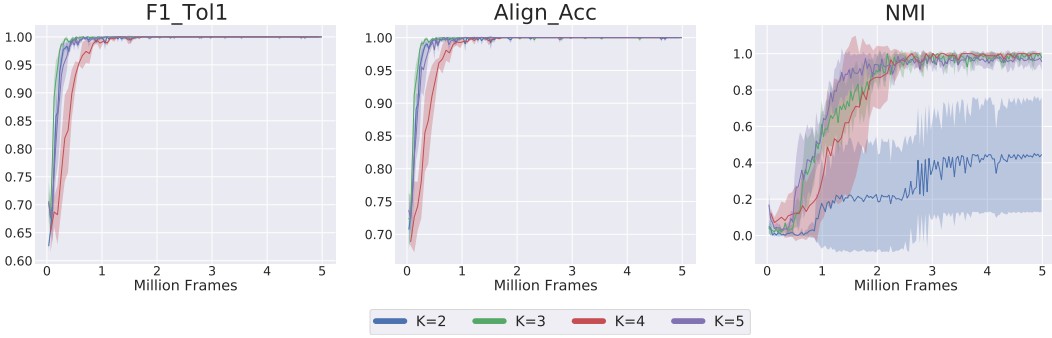

Figure 8: Comparison of parsing results during different $K$ at F1 scores with tolerance, task align accuracy and NMI.

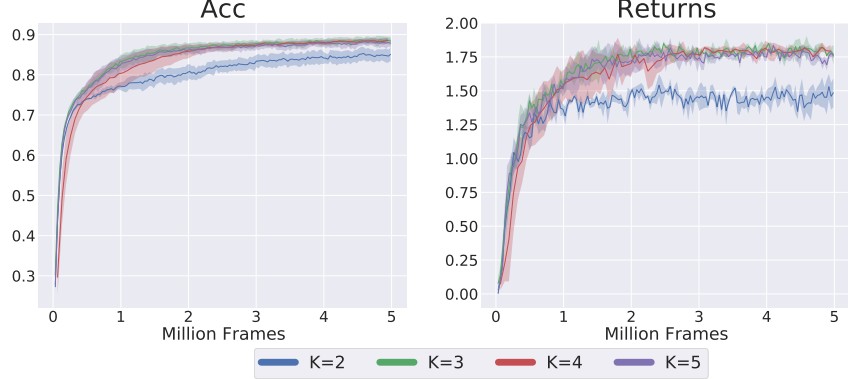

Figure 9: Comparison of prediction accuracy of actions and the returns during different $K$

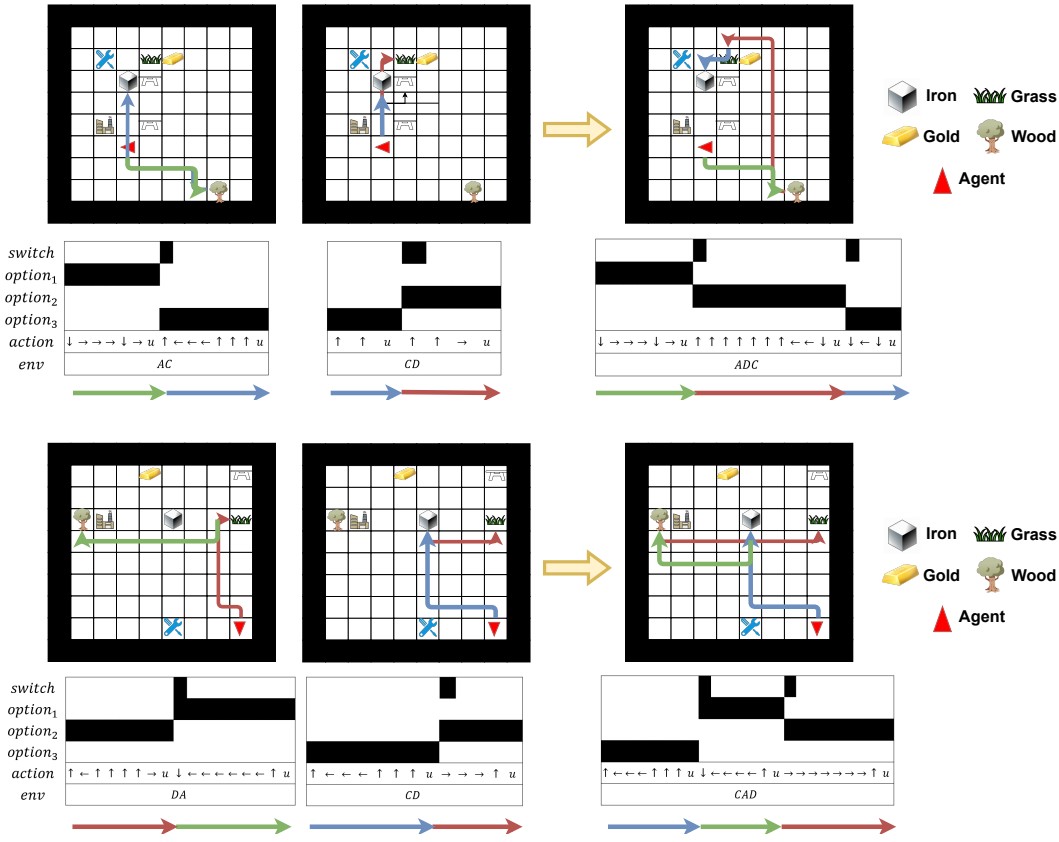

Figure 10: Visualizations of different tasks in **S1**. Different colored lines correspond to different subtasks.

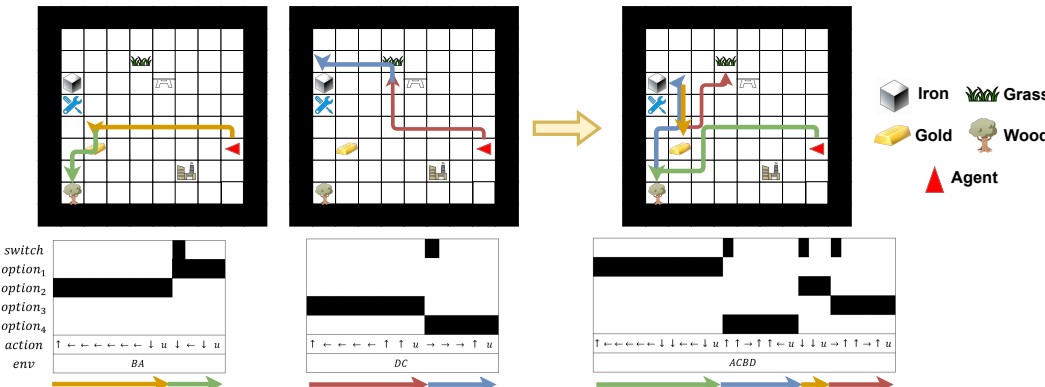

Figure 11: Visualization of a task in **S2**.

## A.6   VISUALIZATION

We show more visualization results in Figure 10 and 11. switch indicates what time the previous option ends and switches to the new option. The options distribution is computed with $\mathbf{p}_t^c$. In Figure 10, the model is trained on different tasks to learn skills(A,C,D) which can be reused in the new tasks(ADC,CAD). In Figure 11, there are two separate OCN models trained on two disjoints tasks sets(AB,BA,CD,DC). The four options can be obtained and reused to solve a new task(ACBD).

