# OpenReview forum: "Inducing Reusable Skills From Demonstrations with Option-Controller Network"
_ICLR.cc/2022/Conference — ICLR 2022 Submitted_

### Official Review · Reviewer_TfQH · 2021-10-30

**Correctness:** 3
**Technical Novelty And Significance:** 2
**Empirical Novelty And Significance:** 3
**Recommendation:** 3
**Confidence:** 5

**Main Review:**

Strengths:

The model proposed is interesting and builds on the recently proposed OMPN model as well as prior work [1, 2, 3, 4, 5] on incorporating inductive biases in RNNs to model temporal hierarchy. Baseline models compared against (CompILE, OMPN, MoE) are strong, appropriate and relevant. All models have been evaluated on the benchmark datasets such as Craft and Dial as in prior work. The pretrained options learned by OCN during the imitation phase lead to good performance gains when being reused later by the controller network. Implementation details for all models (including baselines) have been described well in the Appendix.



Weaknesses:

Assuming I understood the generation of training datasets in experiments S1 and S2 correctly. The training datasets (AC, CD, DA) or (AB, BA) or (CD, DC) generated for the imitation learning phase in S1 & S2 are overly simplistic. The associated decomposition problem involves just finding 1 boundary point to demarcate 2 skill primitives in an episode in the datasets for S1 and S2 experiments. It would be unreasonable to assume this training setup would directly transfer to any general/complex real-world offline datasets containing longer sequences involving execution of many options. Have I missed something in my interpretation? Could the authors clarify this point and/or provide some justification as to why they would expect their model to show similar performance in the more general case?

It would be beneficial to compare and discuss the OCN model in the context of prior work on RNN variants that model temporal hierarchy (Clockwork RNN[1], HM-RNN[2]) using various adaptations to hidden-state updates. Specifically, the HM-RNN update rule uses similar operators such as copy, update (and flush) to allow for slower updates to memory cells deeper in the hierarchy. Therefore it is important to delineate the differences in the update rules employed in HM-RNN [2] and the proposed OCN model to evaluate the novelty/originality of the contribution. Further, earlier work in literature on inducing hierarchical temporal structure in RNNs [3, 4, 5, 6] has not been referenced and compared/discussed in the related work.

Can the hierarchical update rule (control flow between controller and option networks) proposed in the OCN model not be implemented/computed by an appropriately sized OMPN model? Or perhaps a HM-RNN [2] model?



Results:

I found it rather surprising to see that the OMPN seems to struggle to learn good decompositions of primitive skills in S1 which can be composed (reused) by the controller later. Several experiments show CompILE and MoE baselines outperforming the OMPN model quite significantly (Figure 4 and Figure 6). This is very surprising to me since OMPN essentially has a strong inductive bias for modelling temporal hierarchy which is clearly beneficial for these tasks and very similar to the proposed OCN model. Could the authors offer an explanation for why this is so?

It would be interesting to quantify how redundant (i.e. number of options learned for the same underlying skill) the learned options are and amount of reuse by the controller as we use more noisier/complex demonstration datasets for the imitation learning phase.

The authors claim that OCN does not require specifying the exact number of segments in an episode and only need to specify a safe upper bound on it. What do the “extra” slots capture/model? Are they left unused or does it lead to a slightly over-segmented decomposition of the episode?



Writing/Presentation:

In general, the paper contains several typos and ambiguous or unclear phrasing in several sections. The papers’ readability and clarity would greatly benefit from a significant revision in this regard. I have highlighted some of the typos/inconsistent math notation/grammatical errors below:

“model each skill with separate options” -> does this imply that there are several options used to model a single latent skill?

“is a hallmark in human intelligence” -> “is a hallmark of human intelligence”. Further, the authors do not provide a reference to validate the claim in this sentence.

The authors introduce It is unclear whether this is a new problem introduced by the authors or an interpretation of an existing problem. It would help to cite relevant references that define/study this “exploration  problem” in pure HRL methods.

“.. limits the practical values of these approaches” -> “ … limits the practical utility of these approaches”

“ popularity of neural nets” -> “popularity of neural networks”

“However assuming access to an environment in the pertain phase might be infeasible in many tasks” -> missing citation to relevant environments?

“and performs imitation learning” -> “and perform imitation learning”

“our work focused on” -> “our work focuses on”

“performs IRL on the demonstration” -> “performs IRL on the demonstration data”; what does the abbreviation IRL stand for?

“extracts meaning segments” -> “extracts meaningful segments”

“temporal hierarchical structure” -> “hierarchical temporal structure”

“Following the fast and slow learning idea proposed in Madan et. al 2021“ Please improve the citation, some suggestions for the same [6, 7]


Some of the math notation in the “Methodology” section is confusing, difficult (unintuitive) to follow for the reader and also not consistent throughout the text. Using boldface for functions (i.e. RNN updates, example $o_i (x_t, h_{t-1}), c (x_t, h_{t-1})$) is confusing as a small letters with bold font is being used to denote vectors as well. Typo in equation 3? Should it be $\hat{h_{t-1}}$ and not $h_{t-1}$? In Equations (3,4,5,6) several vectors with no bold font. It is also very confusing to have both $h_t$ and $\hat{h_t}$ in the cell network equations. Further, the indexing notation (i, t) for activations of option models (ex: $h_{i, t}^{o}$) is also very confusing and reduces the clarity and readability. Please use another indexing scheme for activations of option models like $h_t^{o_i}$ or something similar. It is also incredibly hard to mentally keep track of 3 variants of each activation vector (for ex: $\hat{p}$, $p’$ and $p$) throughout the description of the algorithm and their roles. Please use a more intuitive alternative to make the reading experience better and improve the clarity. The objective function to minimise (equation 18) is written in a pseudocode-like manner. Please remain consistent and define the objective function using a mathematical expression to avoid ambiguity.


[1] Koutnik et. al, “A Clockwork RNN.”, ICML 2014.

[2] Chung et. al, “Hierarchical Multiscale Recurrent Neural Networks”, ICLR 2017.

[3] Schmidhuber et. al, "Learning Complex, Extended Sequences Using the Principle of History Compression,", Neural Computation 1992,

[4] Mozer et. al,  "Induction of multiscale temporal structure.", NIPS 1992.

[5] El Hihi et. al, “Hierarchical recurrent neural networks for long-term dependencies.”, NIPS 1995.

[6] Jaynes, Edwin T. (1957). Information Theory and Statistical Mechanics. II. Physical Review 108 (2):171.

[7] Boltzmann, Ludwig,  "Studien über das Gleichgewicht der lebendigen Kraft zwischen bewegten materiellen Punkten", Wiener Berichte. 58: 517–560, 1868.


**Summary Of The Paper:**

The paper proposes a new model called Option-Controller Network with requisite inductive biases to model temporal hierarchy. This model is used to learn temporal abstractions and the control policy in the space of options using demonstration data via imitation learning. The performance on discrete action (Craft) and continuous action (Dial) environments are promising compared to existing baselines such as OMPN, CompILE and MoE.

**Summary Of The Review:**

The proposed OCN model is an interesting extension to the recent OMPN. However, my main issue is that the demonstration dataset generation process has drastically reduced the full sequence decomposition task to an overly simplistic case. The writing can be improved significantly as suggested above to improve clarity of the proposed ideas. Further, some closely related prior work [2] on modelling temporal hierarchy in RNNs using similar ideas have not been compared/discussed and many others not referenced [1, 3, 4, 5].

---

> ### Comment · Reviewer_TfQH · 2021-11-29
> **Rebuttal missing**
>
> Given the lack of author response during the rebuttal phase to address the reviewers concerns. I will stick to my initial score of 3 and recommend to reject the paper. I hope the authors use feedback/suggestions provided to improve the paper for a future venue.

---

### Official Review · Reviewer_2XVq · 2021-11-02

**Correctness:** 3
**Technical Novelty And Significance:** 1
**Empirical Novelty And Significance:** 2
**Recommendation:** 3
**Confidence:** 4

**Main Review:**

The overall ideas of task decomposition and using information from demonstrations to accelerate RL are important and this paper proposes a new architecture to address these ideas. It is overall well written and easy to follow. However, the particular network is quite similar to existing implementations and has contributions focused on an aspect that could fall under implementation details in other papers. A further big challenge is the limited evaluation and a missing connection to much of related work. In its current form, the submission is well worth discussion but fits more in the context of a workshop.

Regarding the evaluation it is unclear why the authors do not compare to using options from known approaches like DDO (cited in paper), Option Critic [1] or HO2 [2] (where the latter two can naturally use BC pretrained options as it just requires a change of the initial parameters). The evaluation uses variations of two domains which are not compared against methods from the papers which propose them. In addition, one of the baselines, OMPN, uses an adaptation of the Craft domain in the paper but shows a very different level of performance which suggests that the variation of the domain is different for the submitted paper. The original reason for investigating the papers was that all baselines flatline at surprisingly low performance (most prominent in the Craft domain).

On the more positive side: Section 4.1 includes an example from transferring options from multiple previous experiments which provides an interesting perspective. A future iteration could benefit from more focus on this generally under-investigated idea. But the results for baselines are again surprisingly low.


Minor:
Section 1: the mentioned exploration problem is more a property of specific domains rather than algorithms
‘High level controller is updated less frequently’ is likely supposed to mean acting. Updating can be confused with learning.
‘each option does not correspond to a meaningful subtask’ It is unclear what this is supposed to mean. What is a ‘meaningful subtask’? And why is it good for options to not correspond to one?

[1] Bacon, Pierre-Luc, Jean Harb, and Doina Precup. "The option-critic architecture." Proceedings of the AAAI Conference on Artificial Intelligence. Vol. 31. No. 1. 2017.
[2] Wulfmeier, Markus, et al. "Data-efficient hindsight off-policy option learning." International Conference on Machine Learning. PMLR, 2021.



**Summary Of The Paper:**

The paper proposes a specific neural network architecture for option learning with recurrency on both option and high-level controller. The final action outputs are determined from a mixture of experts of all options. The approach learns options offline from demonstrations (behavioural cloning) and combines them online by learning a new high-level controller via RL (PPO) while using frozen options. The submission uses variations of 2 domains from prior work for evaluation and shows good performance.


**Summary Of The Review:**

Submission proposes a specific architecture for option learning. An aspect that in other work might fall under implementation details. The work has some interesting experiments on combining options from multiple experiments, but the evaluation and connection to previous work is very limited.

---

> ### Comment · Reviewer_2XVq · 2021-11-24
> **Rebuttal missing.**
>
> Giving overall negative scores across reviewers and missing rebuttal, I remain in recommending rejection to provide the authors with additional time to improve the submission.

---

### Official Review · Reviewer_RMbd · 2021-11-04

**Correctness:** 3
**Technical Novelty And Significance:** 3
**Empirical Novelty And Significance:** 3
**Recommendation:** 5
**Confidence:** 5

**Main Review:**

### Strengths

- The proposed approach is intuitive and easy to implement.
- The paper is easy to follow and clearly written.
- The experimental results in CRAFT show significant improvement over prior works.


### Weaknesses

- Comparison to prior latent skill learning approaches is required to understand the contribution of the proposed option extraction method. These approaches already have shown impressive results on complex robotic manipulation environments. It is unclear whether the proposed method can achieve better performance compared to these approaches.
	- Pertsch et al. Accelerating Reinforcement Learning with Learned Skill Priors, CoRL 2020
	- Shankar et al. Learning Robot Skills with Temporal Variational Inference, ICML 2020
- In the approach section, "OCN can be easily expanded to a multi-level model." does not seem true. Most hierarchical approaches so far have failed to learn more than two-level hierarchy, which implies challenges in extending to multi-level hierarchy.
- When an option is continued, the hidden state of the controller is copied without any update. This results in missing task-specific information during the option execution. Given the fact that the proposed method relies heavily on recurrent models, this missing temporal information could make the agent decision not optimal.
- There are discrepancies between pre-training and training OCN models, where the pre-training stage computes an action and termination signal from all options with soft weighting while the RL training stage chooses a single option at a time. Will this affect RL training badly? The options learn to be mixed together during skill learning without clear skill decomposition. But, when using these options, they need to work alone and have a clear skill boundary.
- In the Dial task, the proposed method works better than the baseline methods. However, the variance in Figure 6 seems very high and the performance does not look significant. How many random seeds are used for the experiments? It definitely requires more random seeds to reach a meaningful conclusion. Another question from the learning curves is about the discrepancy between dense and sparse reward settings. OCN seems to succeed 20-30% in the sparse reward setting, but in the dense reward setting, it's not even close to solving 2 subtasks out of 3 subtasks. Moreover, this poor performance makes the scalability of the proposed method less convincing. Can OCN even learn the tasks it saw during pre-training (such as [1, 2])?
- In the Dial task, the experiments focus on generalizing to new tasks with a longer sequence of subtasks. One interesting experiment would be learning skills from a large number of tasks (e.g. three digits pressing tasks with all numbers) and then learning a controller for a holdout task.



### Minor comments

- "Option-Control Network" is used instead of "Option-Controller Network" in the introduction.
- "option framework" -> "options framework"
- "furthered combined" -> "further combined"
- In equation (13), LHS should be $\hat{p}^a_{t}$, not $\hat{p}^a_{i, t}$.
- What happen if the chosen option outputs "done"?


**Summary Of The Paper:**

This paper proposes to extract options from a dataset using an options framework with a recurrent controller and multiple recurrent option networks. During pre-training (i.e. skill extraction), the actions from the framework are computed by the weighted sum over options, which can be end-to-end differentiable, and trained using behavioral cloning. Once all options are extracted from the dataset, a new controller can be trained to solve a new task with the learned and frozen options via RL. The experimental results on the discrete CRAFT environment are impressive, outperforming all baselines. The results on the continuous DIAL environment show improved results over the baseline methods but the difference is marginal.

**Summary Of The Review:**

This paper tackles an important problem of reusable skill (option) extraction from a large multi-task dataset. The proposed method is simple and works well in the discrete task, CRAFT. However, the training process is not intuitive (soft selection during pre-training and hard selection during RL). The experimental results on the continuous domain, DIAL, are noisy and show only marginal improvement over prior work. Moreover, comparisons to continuous latent skill approaches are required to show the technical contribution of the proposed method.

---

> ### Comment · Reviewer_RMbd · 2021-11-29
> **After rebuttal period**
>
> Given missing author response and reviewers' consensus on insufficient clarity and weak empirical results, I would maintain my original rating, weak rejection.

---

### Official Review · Reviewer_XgEE · 2021-11-05

**Correctness:** 3
**Technical Novelty And Significance:** 2
**Empirical Novelty And Significance:** 2
**Recommendation:** 5
**Confidence:** 4

**Main Review:**


Strengths:
- Although this work extends the paradigm of imitation-finetune, its idea is still novel by freezing the options and fine-tuning the controller.
- The topic is timely important, which is related to transfer learning and multi-tasking.
- The paper is well-organized and its method is sound to me. Besides, the related work is quite clear and includes the relevant literature.


Weakness:
- There lacks the clarity. For example, why "positive rewards at the beginning of training" is a problem in HRL? what are structured exploration and unstructured exploration? "During the execution of an option, if probability $e_{i,t}$ is 0, the option will keep executing", what if the task is really hard and $e_{i,t}$ keeps 0, will this step into an infinite loop? According to Figure 1, Figure 3(e), and Table 1, does that mean each option corresponds to a specific task? If this is the case, there lacks analysis or discussion on the scalability and generalizability.
- This work primarily offers an empirical contribution. However, the empirical evaluation is still weak. Regarding the transferability demonstrated in the experiment, the "newly" task is a similar task in the same domain, which is still limited.

Minor:
- typo: "They encoder skills" -> "They encode skills"


**Summary Of The Paper:**


This work targets the issues in learning from demonstration, in order to achieve the ability of skill reuse/ transferability and improve the sample efficiency. To address the issues, the authors propose an option-control network (OCN), including a high-level controller and a pool of low-level options, which is an "imitation-finetune" paradigm as well. Finally, the proposed method is evaluated on two domains, Craft (discrete action space) and Dial (continuous action space), with results demonstrating its effectiveness.

**Summary Of The Review:**

I prefer to weak reject ( marginally below the acceptance threshold) the paper.
- The clarity is an issue that makes it a little bit hard to follow.
- The empirical evaluation is weak so that the transferability looks limited. The scalability and generalizability should be discussed in the paper as well.

---

> ### Comment · Reviewer_XgEE · 2021-11-29
> **Rebuttal missing**
>
> Since there lacks of the author's response, I will stick to my initial score of 5. I hope the authors improve the paper for a future venue.

---

### Decision · Program_Chairs · 2022-01-20

**Decision:**

Reject

**Comment:**

Description of paper content:

The paper describes a technique to learn option policies using behavioral cloning and then recombine them using a high-level controller trained by RL. The underlying options are frozen. The method is tested in two published environments: a discrete grid world environment and a continuous action space robot. It is compared to three baselines.

Summary of paper discussion:

All reviewers moved to reject based on a lack of novelty and a lack of significant empirical results. No rebuttals were provided.